# The flipped classroom is effective for medical students to improve deep tendon reflex examination skills: A mixed-method study

**Shun Uchida** [ID]*, **Kiyoshi Shikino** [ID]°, **Kosuke Ishizuka** [ID]°, **Yosuke Yamauchi**[‡], **Yasutaka Yanagita** [ID][‡], **Daiki Yokokawa** [ID][‡], **Tomoko Tsukamoto**[‡], **Kazutaka Noda**[‡], **Takanori Uehara** [ID][‡], **Masatomi Ikusaka**[‡]

Department of General Medicine, Chiba University Hospital, Chiba, Japan

° These authors contributed equally to this work.
‡ These authors also contributed equally to this work.
* shun.uchida@chiba-u.jp

## Abstract

Deep tendon reflexes (DTR) are a prerequisite skill in clinical clerkships. However, many medical students are not confident in their technique and need to be effectively trained. We evaluated the effectiveness of a flipped classroom for teaching DTR skills. We recruited 83 fifth-year medical students who participated in a clinical clerkship at the Department of General Medicine, Chiba University Hospital, from November 2018 to July 2019. They were allocated to the flipped classroom technique (intervention group, n = 39) or the traditional technique instruction group (control group, n = 44). Before procedural teaching, while the intervention group learned about DTR by e-learning, the control group did so face-to-face. A 5-point Likert scale was used to evaluate self-confidence in DTR examination before and after the procedural teaching (1 = no confidence, 5 = confidence). We evaluated the mastery of techniques after procedural teaching using the Direct Observation of Procedural Skills (DOPS). Unpaired t-test was used to analyze the difference between the two groups on the 5-point Likert scale and DOPS. We assessed self-confidence in DTR examination before and after procedural teaching using a free description questionnaire in the two groups. Additionally, in the intervention group, focus group interviews (FGI) (7 groups, n = 39) were conducted to assess the effectiveness of the flipped classroom after procedural teaching. Pre-test self-confidence in the DTR examination was significantly higher in the intervention group than in the control group (2.8 vs. 2.3, P = 0.005). Post-test self-confidence in the DTR examination was not significantly different between the two groups (3.9 vs. 4.1, P = 0.31), and so was mastery (4.3 vs. 4.1, P = 0.68). The questionnaires before the procedural teaching revealed themes common to the two groups, including "lack of knowledge" and "lack of self-confidence." Themes about prior learning, including "acquisition of knowledge" and "promoting understanding," were specific in the intervention group. The FGI revealed themes including "application of knowledge," "improvement in DTR technique," and "increased self-confidence." Based on these results, teaching DTR skills to medical students in flipped classrooms improves

**Data Availability Statement:** All relevant data are within the paper and its Supporting information files.

**Funding:** The authors received no specific funding for this work.

**Competing interests:** The authors have declared that no competing interests exist.

readiness for learning and increases self-confidence in performing the procedure at a point before procedural teaching.

## Introduction

The deep tendon reflex (DTR) examination is an essential physical examination skill that is assessed in the objective structured clinical examination (OSCE) before clinical clerkships. Even after the OSCE, many medical students are not confident in their skills despite understanding the need and importance of neurological examination. Possible reasons are lack of confidence in the knowledge and performance of the procedure and interpretation of the findings [1]. Inappropriate or inadequate physical examination, including DTR, can result in avoidable adverse events [2]. Thus, effective education about DTR is needed in parallel with a clinical clerkship.

In this study, the flipped classroom was adopted as a teaching technique. In traditional classes, students are taught in class first. Subsequently, they review the learnings and do the homework. In a flipped classroom, students need to learn before attending the classroom [3]. The flipped classroom emphasizes knowledge utilization through problem-solving and is more effective than traditional classes that accentuate knowledge provision [4]. Based on a revised version of Bloom's taxonomy in cognitive domains [5], we can assign the mastery of procedural knowledge about DTR not only to the level of "remember" but also to a higher level such as "understand" or "apply." We planned to use an e-learning video as material for prior learning. Using video materials has been known to be effective in teaching physical examination [6,7].

There have been no reports about the usefulness of flipped classrooms in teaching DTR in the literature. We evaluated the usefulness of a flipped classroom compared with a traditional classroom in a mixed-method study by integrating quantitative and qualitative assessments. We used a 5-point Likert scale to quantitatively evaluate self-confidence in DTR examination before and after the procedural teaching. In addition, we evaluated mastery of techniques after procedural teaching using the Direct Observation of Procedural Skills (DOPS). DOPS is a method of directly observing and assessing techniques performed by learners, and it is frequently used and validated as a method of technique assessment [8]. Learners are expected to acquire higher-order intellectual skills. These skills are difficult to evaluate quantitatively; thus, we assessed learners qualitatively.

## Materials and methods

### Ethics statement

This research was performed in accordance with the Declaration of Helsinki and approved by the Ethics Review Committee of the Chiba University Graduate School of Medicine (Chiba, Japan). The researchers explained to the participants and verbally obtained their informed and voluntary consent. As the researchers took charge of the faculty members, the conflict of interest could arise between the researchers and the participants (the students). As a countermeasure, the faculty members explained to the students that the study would not be used for university grading.

### Trial design

The flipped classroom was conducted in the intervention group, and the traditional face-to-face classroom was conducted in the control group. The two groups were compared to

examine the usefulness of the flipped classroom. The learning objectives of the class were set participants should be able to derive DTR in five areas: biceps, triceps, flexor brachii, patellar tendon, and Achilles tendon. In the clinical clerkship, participants were being rotated in groups assigned by the university, before this study began. The groups of participants were allocated to the intervention or control groups. To integrate quantitative and qualitative assessments, we designed a mixed-method study. In the intervention group, a focus group interview was planned for each rotating group as an additional qualitative assessment. This study was conducted based on the CONSORT 2010 guidelines [9] (S1 Fig).

## Participants

A total of 83 medical students participated in the study. They were fifth-year students who participated in a clinical clerkship at the Department of General Medicine, Chiba University Hospital, from November 2018 to July 2019. The participants were part of 124 fifth-year students and all students who were not yet practicing at the department. This study was also a part of the clinical clerkship education in the department; thus, the participants were not sampled randomly. In this clinical clerkship, students were rotated in groups, each comprising five to six students. All participants learned about basic physical examinations, including DTR in fourth-year classes, and passed a pre-clinical OSCE in which those skills were assessed. The participants' age, sex, and completion of a clinical clerkship in neurology were collected as demographics.

## Intervention

In the intervention group, participants learned about DTR by seven minutes of e-learning, followed by undergoing ten minutes of procedural teaching. In the control group, participants attended face-to-face classes about DTR and then underwent ten minutes of procedural teaching (Fig 1). A video available on this website was used as the e-learning material [10]. This video describes DTR, the anatomy of tendons, the locations to blow, and the posture of the examinee. Medical students in the intervention group were able to watch the video repeatedly on their smartphone, tablet, and PC. In the control group, participants underwent face-to-face learning about DTR instead of e-learning. In the interest of fairness, the video was made available for students of the control group after this study.

Three faculty members participated in this study, of which one faculty conducted procedural teaching. The instructional design was conducted considering the opinions of all faculty members, and teaching skills were standardized before the study began.

## Outcome measures

Self-confidence in DTR examination and mastery of the techniques were compared between the two groups. After the procedural teaching, post-test self-confidence was scored on a 5-point Likert scale, and mastery of the techniques was evaluated using DOPS. As an adjunct, pre-test self-confidence before the procedural teaching was scored. That was scored after prior learning in the intervention group and before face-to-face teaching in the control group (Fig 1).

After the procedural teaching, the evaluators directly observed and evaluated the procedures performed on a simulated patient who was played by another student. The item, "technical ability," of DOPS was analyzed. This item evaluates a mastery on a scale of 1 to 6. The inter-observer reliability could not be assessed by statistical method because research resources did not allow evaluation by multiple observers for a single participant. Instead, the scaling criteria were defined through faculty development to improve inter-observer reliability. The

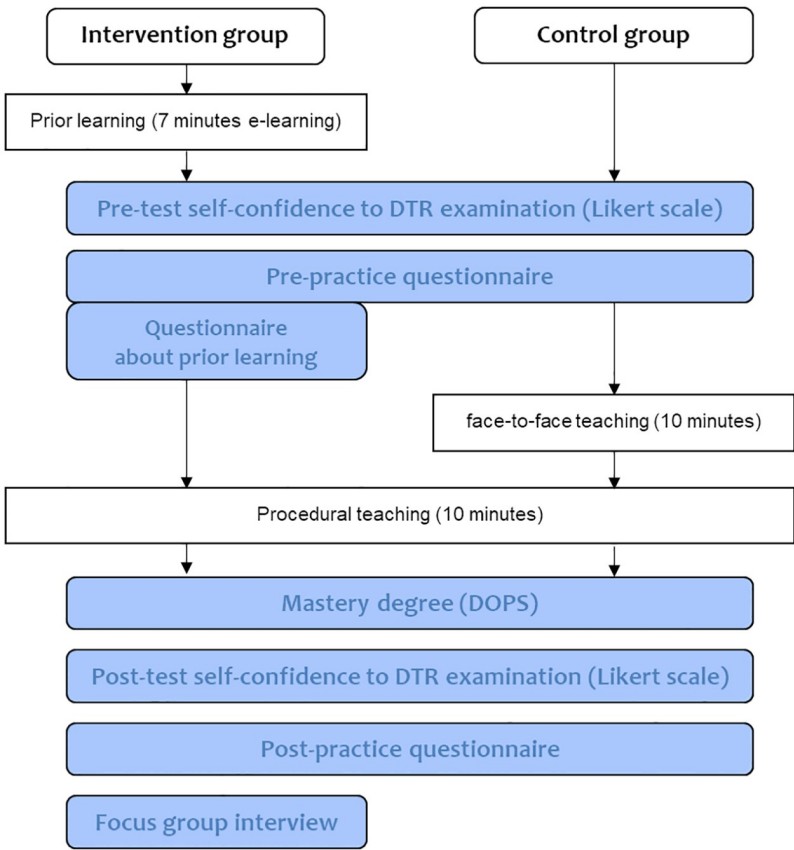

**Fig 1. Research flow.**

scaling criteria were defined as follows: a score of 3 means the borderline student can derive part of DTR; a score of 4 implies the student can derive all DTR; a score of 2 or less indicates below expectation; and a score of 5 or more means above expectation. To ascertain the baseline, the participants' age, sex, and completion of a clinical clerkship in neurology were investigated. In the intervention group, "accessibility of the prior learning material" was investigated on a 5-point Likert scale for reference to the material's validity.

## Sample size

All fifth-year students who were not yet practicing at the Department of General Medicine participated because this study was also a part of clinical clerkship education in the department. A total of 83 medical students participated in the study. The required sample size was 26 in each group, for a two-tailed test of the difference in means between the two groups, assuming a significance level of 0.05, a statistical power of 0.8, and an effect size of 0.8. The sample size was calculated using G*Power (Faul, Erdfelder, Lang, and Buchner, 2007) [11].

## Randomization

The 15 groups were allocated to the intervention or control groups. The groups were allocated through block randomization using Microsoft Excel (Microsoft Co., Redmond, WA, USA) to

the intervention and control groups. The allocation was not blinded to the participants or the evaluators. The same person participated as a faculty member and an evaluator.

### Statistical method

All statistical analyses regarding the 5-point Likert scale and DOPS were performed using SPSS Statistics for Windows 26.0 (IBM Co., Armonk, NY, USA), setting the significance criterion at 0.05. An unpaired t-test was used to analyze the differences between the two groups. A paired t-test was used to analyze differences in pre-test and post-test within each group.

**Free description questionnaire.** A free description questionnaire was used to qualitatively assess self-confidence in DTR examination before and after procedural teaching. These evaluations were conducted using the same questionnaires with the evaluation of self-confidence. The free description questionnaire was expected to provide explanations for the quantitative results. Opinions about prior learning or flipped classrooms were also collected in the intervention group.

**Focus group interview.** In the intervention group, focus group interviews (FGI) [12–14] were conducted for a qualitative assessment. Considering the study objective, we set the theme to assess the effectiveness of the flipped classroom after procedural teaching. Because of the set of this theme, the FGI was conducted only in the intervention group. Three evaluators (SU, KS, and KI) took charge as interviewers. The interview guide was prepared after a discussion with the three interviewers (S1 Table). All students in the intervention group (n = 39, seven groups) were interviewed. In the interview, the faculty member interviewed the group that consisted of five to six students after the procedural teaching. The interviewer asked the following question: "Think of the advantage of the flipped classroom. Why do you feel that it was an advantage?" To analyze the interviews, the verbatim reports were created based on the content recorded on a digital voice recorder. These reports were analyzed qualitatively by content analysis. The two researchers (SU, KS) performed the analysis and consensus-building. We performed researcher triangulation to ensure the quality of the analysis. Cohen's kappa coefficient was used to assess inter-rater reliability [15].

### Mixed methods research

To integrate quantitative and qualitative assessments, a mixed methods study was designed as an exploratory sequential design [16]. The qualitative assessment was intended to support and explain the results of the quantitative assessment as explanations.

## Results

The 15 groups were allocated to the intervention group (7 groups) or the control group (8 groups). As a result, all participants (n = 83) were assigned to the intervention group (n = 39) or the control group (n = 44). There was no statistical difference between the two groups in terms of age, sex, and completion of a clinical clerkship in neurology. Each item was analyzed by an unpaired t-test and resulted in the following *P*-values: *P* = 0.99, *P* = 0.79, *P* = 0.65. In all participants (n = 83), the mean (SD) age was 23.1 (2.0) years, 80.7% (n = 67) were male, and the completion of a clinical clerkship accounted for 41.0% (n = 34). In the intervention group (n = 39), the mean age was 23.2 (2.5) years, 79.5% (n = 31) were male, and the completion of a clinical clerkship accounted for 43.6% (n = 17). In the control group (n = 44), the mean age was 23.0 (1.4) years, 81.8% (n = 36) were male, and the completion of a clinical clerkship accounted for 38.6% (n = 17).

Post-test self-confidence values in DTR examination were 3.9 (0.8) and 4.1 (0.7) in the intervention and control groups, respectively, with no significant difference between the two

**Table 1. Pre-test and post-test self-confidence in DTR examinations.**

|  | Pre-test mean (SD)* | Post-test mean (SD)* | t-value | P-value |
|---|---|---|---|---|
| Intervention group (n = 39) | 2.8 (0.8) | 3.9 (0.8) | 7.69 | <0.001 |
| Control group (n = 44) | 2.3 (0.8) | 4.1 (0.9) | 9.98 | <0.001 |

(* 1: Not at all, 2: Not confident, 3: Confident, 4: Very confident, 5: Extremely confident).

**Table 2. Content analysis of pre-practice questionnaire (n = 83).**

| Free Description | Sub-theme | Theme |
|---|---|---|
| "I don't know how to perform the procedure well." | Lack of procedural knowledge | Lack of knowledge |
| "I have no confidence because I have never performed the procedure on a patient." | Lack of self-confidence because of poor experience | Lack of self-confidence |

**Table 3. Content analysis of pre-practice questionnaire about prior learning in the intervention group (n = 39).**

| Free Description | Sub-theme | Theme |
|---|---|---|
| "I've figured out how to perform the procedure well." | Acquisition of procedural knowledge | Acquisition of knowledge |
| "I found the e-learning easy to understand." | Promotion of understanding using video materials | Promotion of understanding |

groups ($P$ = 0.31). The mastery values evaluated by DOPS were 4.3 (0.8) and 4.1 (0.9) in the intervention and control groups, with no significant difference between the two groups ($P$ = 0.68). Self-confidence in the post-test was significantly higher than that in the pre-test in both groups: t = 7.7, P<0.001 in the intervention group and t = 10.0, P<0.001 in the control group (Table 1).

Pre-test self-confidence in DTR examination (1 = no confidence, 5 = confidence) had mean (SD) values of 2.8 (0.8) and 2.3 (0.8) in the intervention and control groups, respectively—significantly higher in the intervention group ($P$ = 0.005).

The free description questionnaires before the procedural teaching revealed themes common to the two groups, including "lack of knowledge" and "lack of self-confidence." Under the theme "lack of self-confidence," the sub-theme "lack of self-confidence because of poor experience" was extracted (Table 2).

The intervention group comprised the following opinions about prior learning: *"I've figured out how to perform the procedure well," "I found the e-learning easy to understand."* From these opinions, "acquisition of knowledge" and "promoting understanding" were extracted as themes. In particular, from "promoting understanding," "promotion of understanding using video materials" was extracted as a sub-theme (Table 3).

The FGI conducted after the procedural teaching in the intervention group (7 groups, n = 39) generated the following opinions: *"I'm glad I had the opportunity to practice the techniques I learned," "It was great to get instruction while practicing the technique,"* and *"I think the technique has become more reliable."* From these opinions, "application of knowledge," "improvement in DTR technique," and "increased self-confidence" were extracted as themes (Table 4).

**Table 4. Content analysis of post-practice FGI* in the intervention group (7 groups, n = 39).**

| Reaction | Sub-theme | Theme |
|---|---|---|
| "I'm glad I had the opportunity to practice the techniques I learned." | Application of procedural knowledge | Application of knowledge |
| "It was great to get instruction while practicing the technique." | Improving DTR techniques through instruction | Improving DTR technique |
| "I think the technique has become more reliable." | Increased self-confidence in DTR technique | Increased self-confidence |

The inter-rater reliability was high (Cohen's kappa coefficient = 0.84). In the intervention group, "accessibility of the prior learning material" was investigated, and the mean score (SD) was 3.7 (0.2) (1 = hard to use, 5 = easy to use).

## Discussion

Teaching DTR skills to medical students in flipped classrooms improves their readiness for learning and increases their self-confidence in performing the procedure at a point before procedural teaching. If the time for procedural teaching is secured, the mastery of DTR will not decline without face-to-face teaching of DTR knowledge, which may lead to further securing of time for procedural teaching.

In the intervention group, pre-test self-confidence in the DTR examination was significantly higher than that in the control group. We extracted the theme "promoting understanding" from the free description questionnaires before the procedural teaching in the intervention group. This theme suggested that the increased self-confidence was a result of sufficient readiness for learning by "understand," which is a precondition of "apply" in Bloom's taxonomy. Readiness for learning is one of the preconditions of self-directed learning and has a significant impact on learning motivation [17]. Readiness for learning tends to be enhanced in the flipped classroom because a learner can use prior learning materials at any time or place. Consequently, the time for active learning in the classroom is secured, and the learner's understanding is deepened [18].

If the time for procedural teaching is secured, the mastery of DTR will not decline without face-to-face teaching about DTR knowledge, which may lead to further securing of time for procedural teaching. There were no significant differences between the two groups in the post-test self-confidence and the mastery of DTR. In the intervention group, prior learning led to securing the time for procedural teaching, which might be why they achieved mastery of DTR equivalent to the control group without face-to-face teaching. This suggests that the flipped classroom could secure more time for procedural teaching by using the time spent on face-to-face teaching.

This study does not reveal the flipped classroom's superiorities in post-test self-confidence or mastery of DTR despite expectations for effective teaching because of two reasons. First, procedural teaching had the same duration between the two groups. Second, the quantitative assessment was evaluated at a lower level of Bloom's taxonomy. In Bloom's taxonomy [5], DOPS evaluated how participants "apply" the procedural knowledge. In a previous study on flipped classrooms, the effectiveness of a flipped classroom was assessed at a higher level of Bloom's taxonomy [19]. The study dealt with face-to-face teaching for medical students and revealed the effectiveness at a higher level such as "analyze." Thus, if we evaluate the performance at the "analyze" or "evaluate" levels, flipped classroom would be significantly effective.

Additionally, the qualitative assessment showed no significant difference in the post-test self-confidence between the two groups. When we analyzed the free description questionnaires before the procedural teaching, the majority reported lack of self-confidence due to poor

experience. In Bloom's taxonomy, this means that the experience to "apply" the procedural knowledge is poor. Thus, securing the time for procedural teaching might increase self-confidence. There was no difference in the post-test self-confidence in the two groups, probably because the duration for procedural teaching was the same. The results of FGI revealed that securing the time for procedural teaching and applying the knowledge contributed to increasing self-confidence and mastery of DTR. Thus, we propose that prior learning could not provide the experience of applying procedural knowledge, but procedural teaching can provide it.

In the investigation of accessibility of the prior learning material, the mean score (SD) was 3.7 (0.2). Thus, prior learning was thought to be conducted without significant problems. From the perspective of learner concentration, the duration of video materials should be less than ten minutes [20]. The adopted e-learning material was seven minutes, which was valid. To conduct similar flipped classroom instruction, it is necessary to ensure the quality of prior learning by adjusting the duration of the e-learning material to less than ten minutes to make them more accessible.

There were four limitations to this study. First, the allocation to the two groups could not be blinded to the participants and the faculty members, and subjective bias may have affected the results. In addition, since the university had assigned the groups before the start of this study, there might have been a bias in these assignments, which may have affected the study results. Second, the rate of prior learning implementation was not measured. At least, there was a significant difference in pre-test self-confidence in the DTR examination, as mentioned above. Third, the inter-observer reliability of DOPS-scored mastery of techniques was not statistically assessed. As noted above, multiple observers could not be provided for one examiner. Fourth, the participants were medical students at a single institution. Additionally, there were more males than females among medical students at this institution (males = 80.7%, n = 67 of 83), which may have affected the study results. This predominance of male medical students is characteristic of Japanese medical schools. Notably, this study's results could be applied to other institutions or residents and interns.

## Conclusions

Teaching DTR skills to medical students in flipped classrooms improves their readiness for learning and increases their self-confidence in performing the procedure at a point before procedural teaching.

## Supporting information

**S1 Fig. CONSORT 2010 flow diagram.**
(TIF)

**S1 Table. Interview guidelines.**
(PDF)

**S2 Table. Data.**
(XLSX)

## Acknowledgments

The authors thank the physicians who participated in the present study.

## Author Contributions

**Conceptualization:** Shun Uchida, Kiyoshi Shikino.

**Data curation:** Shun Uchida, Kiyoshi Shikino, Kosuke Ishizuka.

**Formal analysis:** Shun Uchida, Kiyoshi Shikino.

**Investigation:** Shun Uchida, Kosuke Ishizuka.

**Methodology:** Shun Uchida, Kiyoshi Shikino.

**Project administration:** Shun Uchida, Kiyoshi Shikino.

**Resources:** Kiyoshi Shikino.

**Software:** Kiyoshi Shikino.

**Supervision:** Kiyoshi Shikino.

**Writing – original draft:** Shun Uchida.

**Writing – review & editing:** Kiyoshi Shikino, Kosuke Ishizuka, Yosuke Yamauchi, Yasutaka Yanagita, Daiki Yokokawa, Tomoko Tsukamoto, Kazutaka Noda, Takanori Uehara, Masatomi Ikusaka.

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
