## [Decision Letter · Decision Letter 0]

2 May 2022

PONE-D-21-37240The flipped classroom is effective for medical students to improve deep tendon reflex examination skills: A mixed-method studyPLOS ONE

Dear Dr. Uchida,

Thank you for submitting your manuscript to PLOS ONE. After careful consideration, we feel that it has merit but does not fully meet PLOS ONE’s publication criteria as it currently stands. Therefore, we invite you to submit a revised version of the manuscript that addresses the points raised during the review process. Based on the review results, a major revision has been suggested in the paper before making any final decision. Please kindly revised the manuscript based on the reviewers' comments. 

We look forward to receiving your revised manuscript.

Kind regards,

Di Zou

Academic Editor

PLOS ONE

Journal Requirements:

2. Please provide additional details regarding participant consent. In the Methods section, please ensure that you have specified (1) whether consent was informed and (2) what type you obtained (for instance, written or verbal). If your study included minors, state whether you obtained consent from parents or guardians. If the need for consent was waived by the ethics committee, please include this information.

Reviewers' comments:

Reviewer's Responses to Questions

**Comments to the Author**

1. Is the manuscript technically sound, and do the data support the conclusions?

Reviewer #1: Yes

Reviewer #2: No

2. Has the statistical analysis been performed appropriately and rigorously? 

Reviewer #1: Yes

Reviewer #2: No

3. Have the authors made all data underlying the findings in their manuscript fully available?

Reviewer #1: Yes

Reviewer #2: Yes

4. Is the manuscript presented in an intelligible fashion and written in standard English?

Reviewer #1: Yes

Reviewer #2: Yes

5. Review Comments to the Author

Reviewer #1: Overall well written and well constructed study. Sample demographics should be discussed in the Methods (as should use of SD for statistics, when appropriate). The limitations of the study should also include the male sex bias in the sample demographics.

Reviewer #2: Line 86

The participants were allocated to the intervention or control groups.

Comments: How the participants were divided into experimental and control group. was it randomly or quota method?

One group consisted of five to six students who practiced for two weeks in this department.

Comment: This is not clear authors are talking about control / experimental group or Focus group.

Line 90

Participants: A total of 83 medical students participated in the study. They were fifth-year students who participated in a clinical clerkship at the Department of General Medicine, Chiba University Hospital, from November 2018 to July 2019.

Comment: How the study participants were recruited; by using probably sampling or non-probably method. How many students were in each group, this is not clear.

Line93

One group consisted of five to six students who practiced for two weeks in this department. (Confusing statement to the reader)

Line 132

This sample size was more than 27, which is the required sample size for a two-tailed test of the difference in means between the two groups

Comment: If the population size is 83 then with 95% confidence level and 3 margins of error, the sample size should be 68

Line 137

The 15 groups were allocated to the intervention or control groups. The allocation was not blinded to the participants or the evaluators

Comment: How they are allocated the group this is not clear. If authors say, the allocation was not blinded to the participants, it could be bias that could alter the results.

Looking at table 1

Self-confidence in DTR examinations (Pretest), mean (SD)*

Self-confidence in DTR examinations (Post-test), mean (SD)*

Comment: Then authors must use paired t - test and they should add t value also in the table

Yes if they want to analyze experimental and control group after intervention ( e learning and face to face practice ) [not pre and post ]then they can use unpaired ( Independent ) t test .

Line 151

In the intervention group, focus group interviews (FGI) [11-13] were conducted for a qualitative assessment.

Comment: Why the focus group was chosen, they could chose randomly from both groups.

Remove errors in sampling method, use tests to check normality of the data then if data is normally distributed then use paired t test ( Pre and Posttest for each group ) 2nd option use unpaired t test but just post test data for each group

6. PLOS authors have the option to publish the peer review history of their article (what does this mean?). If published, this will include your full peer review and any attached files.

Reviewer #1: No

Reviewer #2: No

---

## [Author Response · Author response to Decision Letter 0]

14 May 2022

We hope that the revised manuscript incorporates appropriate revisions as responses to the comments. We believe it has been significantly improved over the initial submission. We trust that our manuscript is now eligible for publication in PLOS ONE. Thank you in advance for your kind consideration of our work.

---

## [Decision Letter · Decision Letter 1]

27 May 2022

PONE-D-21-37240R1The flipped classroom is effective for medical students to improve deep tendon reflex examination skills: A mixed-method studyPLOS ONE

Dear Dr. Uchida,

Thank you for submitting your manuscript to PLOS ONE. After careful consideration, we feel that it has merit but does not fully meet PLOS ONE’s publication criteria as it currently stands. Therefore, we invite you to submit a revised version of the manuscript that addresses the points raised during the review process.

We look forward to receiving your revised manuscript.

Kind regards,

Di Zou

Academic Editor

PLOS ONE

Journal Requirements:

Reviewers' comments:

Reviewer's Responses to Questions

**Comments to the Author**

1. If the authors have adequately addressed your comments raised in a previous round of review and you feel that this manuscript is now acceptable for publication, you may indicate that here to bypass the “Comments to the Author” section, enter your conflict of interest statement in the “Confidential to Editor” section, and submit your "Accept" recommendation.

Reviewer #1: All comments have been addressed

Reviewer #2: (No Response)

2. Is the manuscript technically sound, and do the data support the conclusions?

Reviewer #1: Yes

Reviewer #2: No

3. Has the statistical analysis been performed appropriately and rigorously? 

Reviewer #1: Yes

Reviewer #2: No

4. Have the authors made all data underlying the findings in their manuscript fully available?

Reviewer #1: Yes

Reviewer #2: No

5. Is the manuscript presented in an intelligible fashion and written in standard English?

Reviewer #1: Yes

Reviewer #2: Yes

6. Review Comments to the Author

Reviewer #1: The authors have successful revised the paper according to my concerns, hence it is now acceptable for publication.

Reviewer #2: Methods Page 13 Line 128-29

Page 14 Line 136

The faculty development was conducted to improve inter-observer reliability before this study.

Comment: There is no table for this statement authors should add Cohen's kappa values.

Line 146

A total of 83 medical students (15 groups) participated in the study. Each group comprised more than 26 participants

Comment: delete 15 groups it is confusing to the reader. Let ‘s suppose one group comprising 26 participants, then total should be 26 x 15 = 390 participants, it is incorrect because the authors said earlier 83 students of 5th year medical program ….

Line 152

The 15 groups were allocated to ……..

Comment: medical students (n=83) were allocated into experimental (group A =43 or control groups (group B = 39) randomly

Results page Line 188-89

A total of 39 participants (7 groups) were allocated to the intervention group; 44 participants (8 groups) were allocated to the control group.

Comment: authors could say experimental group n= 39 and control group n =44

Table 1 ( authors cannot analyse two groups with different intervention element ) I think I suggested earlier they can choose Posttest values of

7. PLOS authors have the option to publish the peer review history of their article (what does this mean?). If published, this will include your full peer review and any attached files.

Reviewer #1: No

Reviewer #2: No

---

## [Author Response · Author response to Decision Letter 1]

30 May 2022

We hope that the revised manuscript incorporates appropriate revisions as responses to the comments. We believe it has been significantly improved over the initial submission. We trust that our manuscript is now eligible for publication in PLOS ONE. Thank you in advance for your kind consideration of our work.

---

## [Editor Report · Decision Letter 2]

6 Jun 2022

The flipped classroom is effective for medical students to improve deep tendon reflex examination skills: A mixed-method study

PONE-D-21-37240R2

Dear Dr. Uchida,

We’re pleased to inform you that your manuscript has been judged scientifically suitable for publication and will be formally accepted for publication once it meets all outstanding technical requirements.

Kind regards,

Di Zou

Academic Editor

PLOS ONE

---

## [Editor Report · Acceptance letter]

9 Jun 2022

PONE-D-21-37240R2 

The flipped classroom is effective for medical students to improve deep tendon reflex examination skills: A mixed-method study 

Dear Dr. Uchida:

I'm pleased to inform you that your manuscript has been deemed suitable for publication in PLOS ONE. Congratulations! Your manuscript is now with our production department. 

Kind regards, 

on behalf of

Dr. Di Zou 

Academic Editor

PLOS ONE